# No Roots, No Fruits: Marcel Tanner’s Scholarly Contribution, Achievements in Capacity Building, and Impact in Global Health

**DOI:** 10.3390/diseases10040116

**Published:** 2022-12-01

**Authors:** Andrea Graf, Lukas Meier, Jürg Utzinger

**Affiliations:** 1Department of History, University of Basel, 4001 Basel, Switzerland; 2Swiss Tropical and Public Health Institute, 4123 Allschwil, Switzerland; 3University of Basel, 4001 Basel, Switzerland

**Keywords:** capacity building, epidemiology, global health, health systems, infectious diseases, malaria, parasitology, public health, Tanzania

## Abstract

On 1 October 2022, Marcel Tanner celebrated his 70th birthday with his family and friends on the River Rhein in Basel. Trained in epidemiology (Ph.D.) and public health (MPH), Tanner devoted his entire working life to research, teaching, and capacity building. Indeed, he built up productive partnerships, fostered multinational consortia, served on numerous scientific and strategic advisory boards, and contributed measurably to improving people’s health and well-being. We systematically searched the Web of Science Core Collection to identify Tanner’s scholarly contribution and pursued an in-depth analysis of his scientific oeuvre including the main areas of research, pathogens, diseases, and health systems, and the geographical foci of his scholarly activities. Additionally, we examined Tanner’s impact on personal and institutional capacity building in the arena of global health. We also invited a handful of colleagues to describe their experiences while working with Marcel Tanner. What transpires is a considerable breadth and depth of peer-reviewed publications in tropical medicine; epidemiology, environmental, and occupational health; parasitology; and infectious diseases. More than a third of the 622 peer-reviewed articles, the first piece published in 1978, focused on various aspects of the protozoan parasite *Plasmodium* and the disease it causes: malaria. Tanner trained, taught, and inspired generations of students, scientists, and practitioners all over the world. His unique ability to bring people and institutions together to work in partnership is at the heart of an impactful career in global health.

## 1. Introduction

What a great idea to put forth a Festschrift to commemorate Marcel Tanner’s 70th birthday [1]! With pleasure, we took up the invitation and enjoyed the freedom of writing the current piece. Our overriding motivation was to document—quantitatively and qualitatively—Tanner’s contribution to global health. We placed particular emphasis on Tanner’s scholarly contributions in research and complemented our inquiry with his teaching activities and capacity building efforts at individual and institutional levels. We employed different sources of data.

To determine Tanner’s scholarly contributions, we carried out a systematic review using the readily available electronic database Web of Science Core Collection. As regards capacity building, we consulted Tanner’s curriculum vitae and investigated the acknowledgement sections from openly accessible Ph.D. theses put forth by students supervised or co-supervised by Tanner. Additionally, we invited a handful of colleagues who collaborated with Tanner in one way or another and asked them to highlight unique features about his work ethos and collaborative spirit. The data were triangulated to make a comprehensive overview of Tanner’s career.

### Short Biosketch of Marcel Tanner

Marcel Tanner was born on 1 October 1952 and grew up in Basel, Switzerland. As a young student, Tanner was already fascinated by the complex life cycles of parasites and their interactions with human and animal hosts and the environment. He studied biology and graduated with a degree in medical zoology from the University of Basel (1972–1976). Subsequently, he earned a Ph.D. in epidemiology at the Swiss Tropical Institute under the main supervision of Professor Niklaus Weiss. Tanner’s thesis focused on a deeper understanding of the antigenic variation of trypanosomes, the causative agents of human African trypanosomiasis [2].

In 1979, Tanner travelled to Cameroon with his supervisor Professor Weiss to study and treat local populations suffering from onchocerciasis. The stay in Cameroon would later become a turning-point in Tanner’s career. In Cameroon, Tanner observed—firsthand—that medical treatment alone is inadequate to have a long-lasting impact on the health and well-being of neglected populations. Indeed, many people simultaneously suffered from malaria, respiratory tract infections, and diarrhoea, while most of the households lacked access to electricity, clean water, and improved sanitation. This insight led Tanner to acknowledge the vital importance of public health and the need to study diseases in the context of health and social-ecological systems.

In 1981, Tanner became the head of the Swiss Tropical Institute Field Laboratory (STIFL) in Ifakara, Tanzania. Under his leadership and in close partnership with the Swiss Tropical Institute in Basel, STIFL gradually transformed into a centre of excellence in Tanzania, for Tanzania, and led by Tanzanians [3,4,5]. The centre’s major initiatives focused on improving peripheral health systems by deploying and validating primary health care approaches. Particularly noteworthy was a comprehensive research project that analysed the interplay between infection, nutrition, immunity, and the environment [6,7]. Tanner and colleagues were the first to provide evidence that some malaria parasites in Tanzania were resistant to the widely used antimalarial drug chloroquine, a finding for which he earned recognition from the Tanzanian government [8,9]. After handing over the responsibility of STIFL to his successors Dr. Donald de Savigny (1985–1987) and Dr. Christoph Hatz (1987–1988), Tanner returned to Basel and worked as a research scientist at the Swiss Tropical Institute.

In 1986, Tanner enrolled in an MPH course at the London School of Hygiene and Tropical Medicine (LSHTM) in London. Amongst his peers were Dr. Pedro Alonso (who would later become the director of the World Health Organization (WHO) Global Malaria Programme), Dr. Clara Menendez (a global leader in maternal and child health), Dr. Leonardo Simão (who would later become Minister of Health and Minister of Foreign Affairs of Mozambique), and Dr. Max Price (who would later become the Dean of the University of Cape Town in South Africa).

In 1997, Tanner took over the reins of the Swiss Tropical Institute, as the fourth director after Professor Rudolf Geigy (1944–1972), Professor Thierry Freyvogel (1972–1987), and Dr. Antoine Degrémont (1987–1997) [10]. Under Tanner’s directorship (1997–2015), the institute grew from a family-type organisation to one of the world’s leading global health institutions with 715 staff and students from more than 60 nations. He also oversaw the integration of the Institute of Social and Preventive Medicine into the Swiss Tropical Institute, including a name change to Swiss Tropical and Public Health Institute (Swiss TPH) in 2009/2010.

At the University of Basel, Tanner held the chair of medical parasitology and infection biology from 1997–2017. From 2002 to 2004, he served as the dean of the Faculty of Science. In 2006 and 2020, Tanner received honorary doctorates from the University of Neuchâtel and the University of Zurich, respectively.

## 2. Materials and Methods

### 2.1. Systematic Search of Web of Science Core Collection

We determined Marcel Tanner’s scholarly contribution through a systematic search of the Web of Science Core Collection (www.webofscience.com, accessed on 4 September 2022). Our search was performed on 3 August 2022. We included all publications until 31 December 2021 without language restrictions. As we were primarily interested in peer-reviewed articles, we excluded three types of documents: notes, meeting abstracts, and corrections. The remaining entries fell into the following categories: original papers, reviews, editorials, letters, book chapters, and proceedings papers. Some of the records were assigned to two categories simultaneously by Web of Science, in which case they were weighted accordingly (0.5 each).

The publications identified in our search were analysed according to a set of predefined criteria. First, we tabulated the number of publications per year and examined Tanner’s authorship position. Authorship position was stratified as first (in the case of a single-authored piece, the author was considered first author), last (for papers with two authors, the second author was considered last author), or co-author (for all papers with three or more authors, the positions neither at first nor last position were considered co-author). While the number of publications per year could be automatically determined using Web of Science filters, the author position had to be manually evaluated.

Second, the national institutional affiliation of the first authors was analysed. Of note, Web of Science states the institute and country of affiliation of most, but not all, the authors. Whenever the requested information was missing, a manual internet search was performed, including internal consistency checks. Numerous authors have published more than one piece as first author with Tanner. Since these authors were sometimes affiliated with different institutions at the time of the respective publications, their affiliation was evaluated individually for each publication. Additionally, several authors were affiliated with more than one address, sometimes from different countries. In such cases, each affiliation was weighted accordingly (two affiliations were given a value of 0.5 each, three a value of 0.33 each, etc.).

Third, Tanner’s publications were assigned to specific scientific disciplines. Web of Science tags its records according to scientific discipline. Since most publications fall into multiple categories (e.g., a specific publication assigned to both parasitology and tropical medicine), the Web of Science analysis tool first counts the number of records assigned to a category and then calculates the percentage share of each category in the total number of articles. Remarkably, Tanner’s publications were assigned to more than 50 categories. Since this number is too large for a meaningful visualization, we only chose the top eight categories and summed up the remaining ones in a single category labelled “varia”. The choice of the top eight categories was based on an arbitrary truncation at 20 counts within a single category.

### 2.2. Content Analysis

We pursued a content analysis of Tanner’s scholarly contribution, focusing on the main diseases researched and the countries where the studies were conducted. Diseases and pathogens addressed by Tanner’s publications were determined by screening the titles and abstracts for specific keywords (e.g., malaria as disease, *Plasmodium* as pathogen). We defined specific categories for areas that resulted in low frequencies or those that could not be tagged unambiguously to a specific disease category. If the publication focused on two or more diseases or pathogens, these entries were weighted accordingly (0.5 each, 0.33 each, etc.).

To determine the country where Tanner and colleagues pursued their research, the title and abstract of the articles were manually searched for country names or other geographical identifiers. In case two or more countries were evident, each country was weighted according to the total number (i.e., for studies that were conducted in two countries, each country received a value of 0.5, for three countries, each received a value of 0.33, etc.). Those articles that unambiguously stated country-specific study sites were subjected to a deeper analysis. We drew a world map and highlighted the countries where Tanner and colleagues carried out their research.

### 2.3. Impact on Capacity Building

We determined Tanner’s impact on capacity building, by analysing the number of Ph.D. students he supervised or co-supervised. We were particularly interested in the nationalities of the Ph.D. students as a measure of international networks and global outreach. Our analyses draw on Tanner’s curriculum vitae, complemented with a search on the publicly available repository of Ph.D. theses by the University of Basel (see: www.edoc.unibas.ch; accessed on 4 September 2022). In most cases, each student’s nationality is stated on the thesis’ title page. If this information was missing, manual internet searches were conducted in an effort to obtain unambiguous information. Some Ph.D. students indicated dual nationalities, and hence, both countries were assigned a weight of 0.5 each.

Finally, we reached out to eight individuals who collaborated with Tanner in one way or another and invited them to share their personal experience and highlight something special about their work relation with Tanner.

## 3. Results

### 3.1. Number of Publications and Author Status

Figure 1 shows a flow chart outlining the steps of our systematic search of Web of Science Core Collection on 3 August 2022. Overall, there were 712 records for the author “Tanner M”, the large majority of which were affiliated with “Basel”, where Tanner pursued most of his academic career. Corrections, meeting abstracts, and notes were excluded (n = 91), 10 pieces were added through a manual search, and nine articles published in 2022 were removed. Taken together, there were 520 original papers, 43 reviews, 33.5 editorials, 14 letters, 6.5 book chapters, and 5 proceedings papers, summing up to a total of 622 records that were subjected to further analysis.

Figure 2 displays the number of publications per year over the 44-year period from 1978 to 2021, stratified by Tanner’s author position as either first author (n = 38), co-author (n = 394), or last author (n = 190). The first piece, entitled “Studies on *Dipetalonema vitae* (Filarioidea). II. Antibody dependent adhesion of peritoneal exudate cells to microfilariae in vitro” was published in 1978 in *Acta Tropica*, and was co-authored by Professor Niklaus Weiss, Tanner’s Ph.D. supervisor [11]. Of note, *Acta Tropica* was the house journal of the Swiss Tropical Institute from the founding of the institute in 1943 until it was sold to Elsevier in the late 1980s [12].

In the first decade of Tanner’s scientific career (1978–1987), he published a total of 31 articles, usually 1–5 articles per year, with about half of the articles as first author (n = 15). In the following decade (1988–1997), Tanner was associated with 74 articles, mainly as last author. From 1998 onwards, Tanner published 13 articles or more (up to 38 articles in 2015) each year; during this 24-year period, he mainly acted as co-author.

### 3.2. Country of Affiliation of First Authors

Figure 3 shows a world map, highlighting the countries that the first authors of all the publications Tanner co-authored are affiliated with. Despite various search methods, the first author’s affiliation could not be determined for one of the 622 publications, and hence, this publication was removed from the analysis. Overall, first authors were affiliated with 37 different countries. By far the majority of first author affiliations, including Tanner himself who was jointly affiliated with Swiss TPH and the University of Basel, were with an institution in Switzerland (n = 300). The second most common country of affiliation was Tanzania (n = 81). Hence, it is no wonder why Tanner often refers to Tanzania as his second home. The world map reveals a handful of additional countries with a large proportion of first author affiliations, including China (n = 42), Spain (n = 32.7), Côte d’Ivoire (n = 31.2), USA (n = 30.5), and UK (n = 28.8). Australia, where Tanner completed two sabbaticals as a visiting Professor at the University of Queensland in Brisbane in 1996/1997 and 2009/2010, is also highlighted on the map where it accounts for more than 10 of the first author affiliations (n = 14.7). In the remaining 29 countries, the number of first author affiliations was below 10.

### 3.3. Web of Science Core Collection Categories

Tanner’s 622 publications examined here were assigned to 56 categories of scientific disciplines according to Web of Science, with the top eight categories illustrated in Figure 4. First on the list is “tropical medicine” (22%), followed by “public, environmental, and occupational health” and “parasitology” (18% each), and “infectious diseases” (11%). These four categories account for more than two thirds of Tanner’s research output. The categories “medicine general internal”, “immunology”, “microbiology”, and “multidisciplinary sciences” each contributed 2–5%. All other publications, assigned to one or several of the remaining 48 categories, accounted for (18%). The categories cover a wide range of topics from health care sciences to health policy services, medical ethics to social sciences, and pharmacology and medicinal chemistry to environmental sciences, to name but a few.

### 3.4. Main Diseases and Pathogens Studied

Analysing the diseases and pathogens covered in Tanner’s research output, two observations stand out. First, the range of diseases and pathogens studied is impressive. Second, almost two thirds of the publications pertain to parasitic diseases (n = 402.5), with malaria clearly in the lead (n = 243.5) (Figure 5). Malaria is followed by schistosomiasis (n = 106.5), which constitutes the second most important disease researched. Filariasis (n = 13.5, including the keywords lymphatic filariasis and onchocerciasis) and trypanosomiasis (n = 5.5, including Chagas’ disease) are diseases that are covered to a considerably smaller extent in Tanner’s publication output. Under ‘other parasitic diseases’, a range of pathogens and diseases are covered, including the following key words: intestinal parasites, parasitic worms, helminths, geohelminths, soil-transmitted helminths, hookworm, enterobiasis, strongyloidiasis, *Clonorchis sinensis*, *Echinostoma caproni*, *Fasciola hepatica*, *Opisthorchis viverrini*, *Paragonimus westermani*, amoebiasis, giardiasis, and intestinal cryptosporidiasis, *Cryptosporidium*, and leishmaniasis).

In addition to parasitic diseases, publications that focus on viruses (n = 39.5) and bacteria (n = 25) also appeared in Tanner’s research output. About 5% of the publications pertain to HIV/AIDS (n = 29.5). In the past 2 years, a few articles were published about COVID-19 (n = 4.5). Other viral diseases, including norovirus, pegivirus, and Ebola only played a marginal role (n = 5.5) in Tanner’s research output.

As regards bacterial diseases, several publications pertain to tuberculosis (n = 8.5) and other bacterial diseases, including vibrio, *Streptoccocus*, *Staphylococcus*, *Rhodococcus coprophilus*, *Neisseria meningitidis*, *Mycobacterium bovis*, leprosy, cholera, *Coxiella burnetii*, Buruli ulcer, brucellosis, and Q-fever (n = 16.5).

Additionally, 12 publications summarised under the category ‘varia’, cover a range of other diseases, disease symptoms, or health conditions, such as anaemia, goitre, iodine deficiency, hydrocele, fungal infections, encephalitis, diarrhoea, diabetes mellitus, hypertension, hepatitis B, C, and E, meningitis, and sepsis.

Finally, almost a quarter of publications analysed (n = 143) do not cover a disease- or pathogen-specific topic, but rather discuss health care, disease control and elimination, health policy, and health research related topics.

### 3.5. Countries Where Researched Was Pursued

While some of Tanner’s publications, particularly in the earlier years, involve exclusively laboratory research, most of the articles focus on epidemiological studies, clinical trials, and health systems research. Indeed, based on an analysis of article titles and abstracts, 402.4 of the publications could be linked to a specific country where the research was conducted. There were another 19.6 articles that only mentioned broad regions (i.e., Africa, sub-Saharan Africa, West Africa, East Africa, Central West Africa, n = 17.4; Asia, n = 1; South America, n = 1; and Middle East, n = 0.2). The remaining 200 publications could not be linked to any specific geographical setting.

Among the 402.4 study sites, “field work” could be assigned to any of 59 countries. As shown in Figure 6, one country—Tanzania—stands out, where more than half of the research was conducted (n = 212.2). Another four countries where Tanner and colleagues pursued a considerable amount of research emerged: Côte d’Ivoire (n = 49), China (n = 27.5), Switzerland (n = 21), and Chad (n = 13). In the remaining 54 countries, only a few studies were conducted with Tanner’s involvement (less than 10 publications each).

### 3.6. Capacity Building Efforts

During his long career, Tanner supervised and co-supervised numerous MSc, MD, Ph.D., and postdoctoral students. He taught at the Bachelor, Master, and Doctorate level, and was deeply involved in professional postgraduate teaching and training at Swiss TPH, University of Basel, the Swiss School of Public Health (SSPH+), and elsewhere. Moreover, he played an instrumental role in progressing the career of many young scientists towards becoming assistant, associate, and full professors.

According to Tanner’s curriculum vitae, complemented with an online search of a repository of Ph.D. theses by the University of Basel, Tanner supervised 65 Ph.D. students and co-supervised another 235 Ph.D. students. Figure 7 shows the nationality of these 300 doctoral students. More than a third of the Ph.D. students were from Switzerland (n = 112.5), while the remaining students came from 54 nations. Tanzanian students form the second most important cluster (n = 34), followed by students from Germany (n = 23), USA (n = 14), Ghana (n = 10), and Côte d’Ivoire (n = 7). Italy and the UK are represented by 5.5 students each.

### 3.7. Appreciations by Selected Individuals

Six of the eight individuals (former students, collaborators, and peers) who were contacted by e-mail responded to our queries and provided personal statements about their experiences while working with Tanner. Specifically, we were interested in seeing if and how Tanner influenced their work and career developments. Below, we provide the name, country, current position, and personal quote from each respondent.

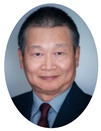
**Jia-Gang Guo**China, National Institute of Parasitic Diseases; and Switzerland, World Health Organization*“In my 40-year career, I was fortunate to meet Professor Marcel Tanner. With his help, I grew up and came closer to international cooperation, step-by-step. He not only gave me a professional degree, but also set a perfect example of a scientist, which was worth emulating for my life.”*(4 September 2022)
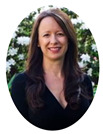
**Jo-An Occhipinti** (neé Atkinson)Australia, University of Sydney*“I am fortunate to call Marcel a mentor and friend. He has long supported women for science leadership and has shaped my career in important ways. Marcel taught me that the advancement of science and public health only comes from having the courage and persistence to chase innovation, rather than ‘metrics and me-too studies’. He also instilled in me the importance of working with ‘head, heart, and hand’.*”(6 September 2022)
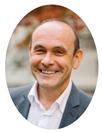
**Mauro Moruzzi**Switzerland, Former Ambassador and Member of the Neuchâtel Municipal Council*“Over the past four decades, Marcel has changed the life of millions of people on five continents, thanks to his incredible, relentless work, both on the field—where he loves to be—and in labs as an outstanding researcher and scientist, in classrooms as a respected and beloved professor, spending half of his time travelling around the world to meet politicians, academics, charity heavy-weights, journalists, civil servants, NGOs representatives, and many more. The secret behind his incredible track-record? He pays every single person, no matter his or her social or economic background, exactly the same respect and he listens carefully to the people’s real concerns, to address the right issues. Your contribution to make this world a more human one, dear Marcel, is an invaluable example and inspiration to all of us, and especially to me: thank you!”*(6 September 2022)
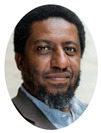
**Salim Abdulla**Tanzania, Ifakara Health Institute*“From a humble beginning, the Ifakara Health Institute has grown to be among the leading health research institutes in Africa. Prof. Marcel Tanner has been the institute’s shining light through this journey, mentoring, guiding, and supporting me and the leading scientists that you see today in the institute. His dedication, humility, kindness to all, commitment to quality work, and perseverance towards solving the pressing health problems in our communities continues to shape the development of the institute and its people.”*(7 September 2022)
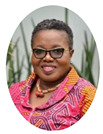
**Margaret Gyapong**Ghana, University of Health and Allied Sciences*“I first met Professor Marcel Tanner in the early 1990s. I was amazed at his intelligence and ability to switch from a highly knowledgeable scientist commenting on updates from parasitology to social science and economics and then to a social person at evening receptions who would crack jokes and make everyone go in stitches with laughter. Little did I know he would become one of my Ph.D. supervisors a few years later. His ability to find time to read my work on a regular basis and give me comments and guidance without fail was remarkable. I owe a lot of my achievements to his guidance and recommendation for which I am truly grateful.”*(7 September 2022)
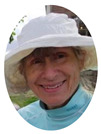
**Joanna Schellenberg**UK, London School of Hygiene and Tropical Medicine*“Marcel, you have more energy and enthusiasm than anyone else I know! You are always ready to identify and engage in the key issues, to solve problems, to connect people, and to share your joy in pushing forward the boundaries of science. And with a wonderful sense of humour! Thank you!”*(5 October 2022)

## 4. Discussion

The bibliometric analysis of Marcel Tanner’s contribution to the peer-reviewed literature, coupled with a quantitative and qualitative appraisal of his role on capacity building, revealed considerable influence and impact in the scientific fields of tropical medicine; epidemiology, environmental, and occupational health; parasitology; and infectious diseases. With his extensive track record, especially in the fields of malaria, schistosomiasis, and health systems research, as well as many decades of highly respected and influential teaching, mentoring, and providing policy advice, Tanner has had a significant impact on global health in many ways.

An early focus of Tanner’s work was the parasitic disease schistosomiasis. Specifically, together with colleagues, Tanner demonstrated the value of portable ultrasonography for morbidity assessment due to *Schistosoma haematobium* at the peripheral level [13]. Subsequently, seminal papers summarised how to monitor and evaluate schistosomiasis control within primary health care programmes [4,14] and how simple school-based questionnaires allow for rapid and cost-effective identification of high-risk communities of urogenital and intestinal schistosomiasis [15,16]. The most widely cited paper in the field of schistosomiasis that Tanner co-authored is a systematic review and meta-analysis of the effect of water resources development and management on disease transmission [17]. Tanner, together with colleagues from Switzerland, China, and Côte d’Ivoire, also significantly contributed to the literature regarding the efficacy and safety of the antimalarial drug artemisinin against schistosomiasis [18,19].

Perhaps Tanner’s most influential contributions to global health stem from his consistent and sustained research on malaria vaccines and other interventions against malaria. In 1992, Tanner and colleagues from Swiss TPH, the Ifakara Health Institute (IHI), and the team of Dr. Pedro Alonso conducted the first phase III clinical trial of a malaria vaccine (SPf66) on the African continent [20,21]. In 2006 and 2008, Tanner was also at the forefront of clinical trials for the successor vaccine, RTS,S, developed by GSK [22,23]. Especially important was a phase II trial which involved 340 children aged 2–4 months in Bagamoyo, Tanzania. The results showed that the vaccine had an efficacy rate of 65.2%, a good safety profile, and was compatible with other childhood vaccines. In 2021, WHO recommended the widespread use of RTS,S for children living in regions with moderate to high *Plasmodium falciparum* malaria transmission [24].

While Tanner welcomed this progress, he insisted that instead of relying on a single “magic bullet”, there is a need to combine different tools and approaches that can be readily tailored to specific social-ecological settings. Research conducted in Tanzania revealed that social marketing of insecticide-treated nets [25,26] and intermittent preventive treatment in infants (IPTi) significantly reduced child mortality [27,28,29].

In addition to his work as a distinguished researcher and teacher, Tanner was involved in several product development partnerships (PDPs), bringing together partners from research, academia, and the private sector. He was a founding member of the *Medicines for Malaria Venture* (MMV) and chaired the *Drugs for Neglected Diseases initiative* (DNDi) and the *Foundation for Innovative New Diagnostics* (FIND), all based in Geneva [30,31].

New tools against neglected tropical diseases and other poverty-related diseases can only be effective within well performing health systems. Tanner contributed substantially to health systems research [32]. According to a ranking conducted in 2014, he is one of the six most productive scholars in health systems research between 1900 and 2012 [33]. For Tanner, however, having impact does not mean having an extensive publication record with high “impact factors”, “h-factors”, and the like. Instead, it means for him to bring together people to solve the most pressing global health challenges and to foster the careers of young scientists from all over the world.

## Figures and Tables

**Figure 1 diseases-10-00116-f001:**
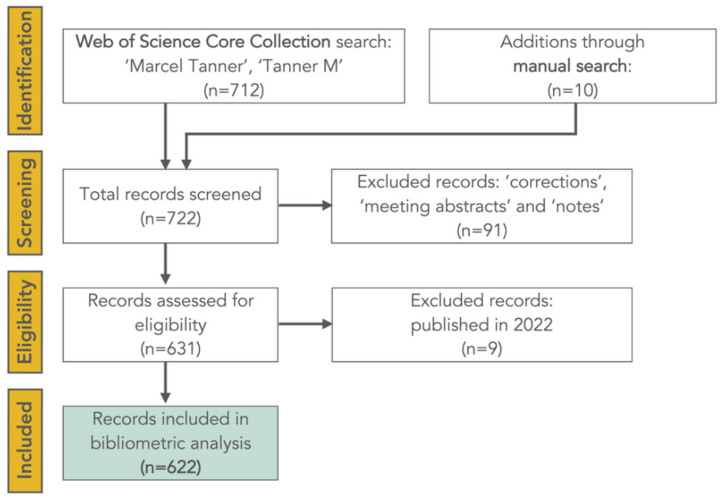
Flow chart of the steps taken to identify Marcel Tanner’s peer-reviewed scholarly contributions between 1978 and 2021 that were subjected to further analyses (source: Web of Science Core Collection, search performed on 3 August 2022).

**Figure 2 diseases-10-00116-f002:**
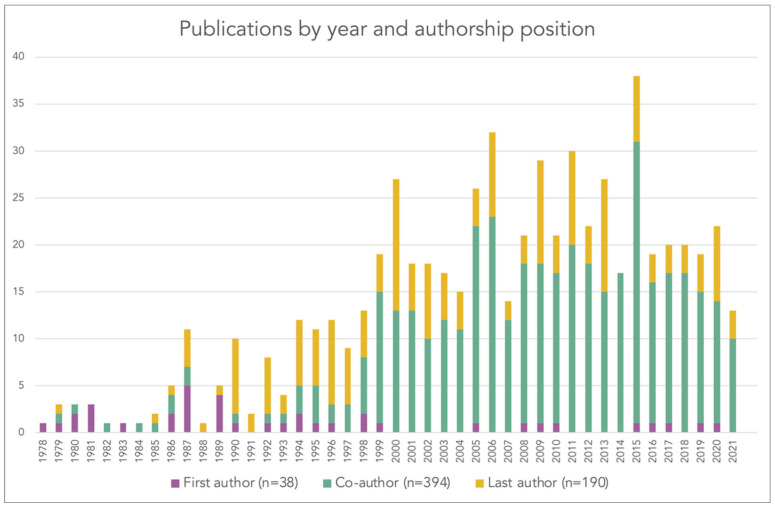
Yearly number of peer-reviewed articles co-authored by Marcel Tanner from 1978 to 2021, stratified by first, co-, and last author (source: Web of Science Core Collection, search performed on 3 August 2022).

**Figure 3 diseases-10-00116-f003:**
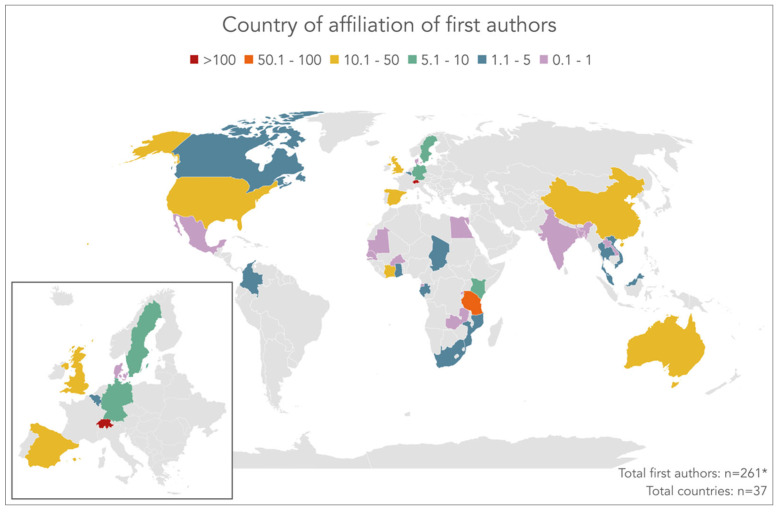
Country affiliation of first authors in peer-reviewed articles co-authored by Marcel Tanner between 1978 and 2021. * For one first author, country affiliation could not be determined (source: Web of Science Core Collection, search performed on 3 August 2022, complemented with internet searches in case of missing information).

**Figure 4 diseases-10-00116-f004:**
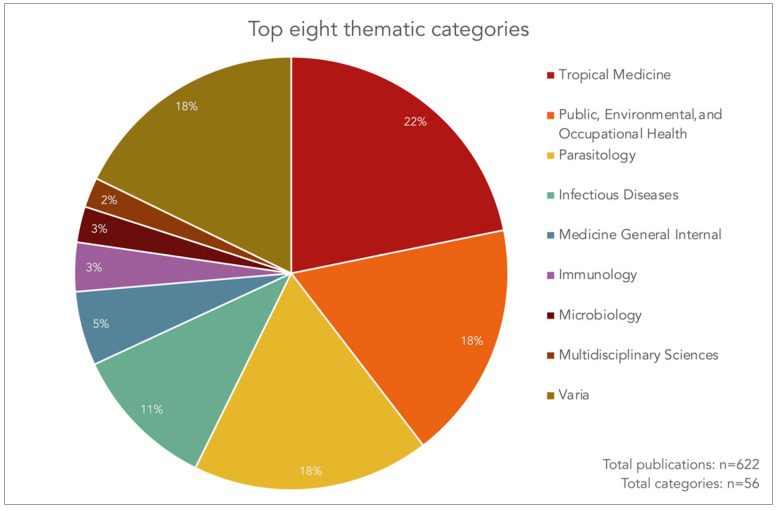
The top eight Web of Science categories for Marcel Tanner’s publications between 1978 and 2021 (source: Web of Science Core Collection, search performed on 3 August 2022).

**Figure 5 diseases-10-00116-f005:**
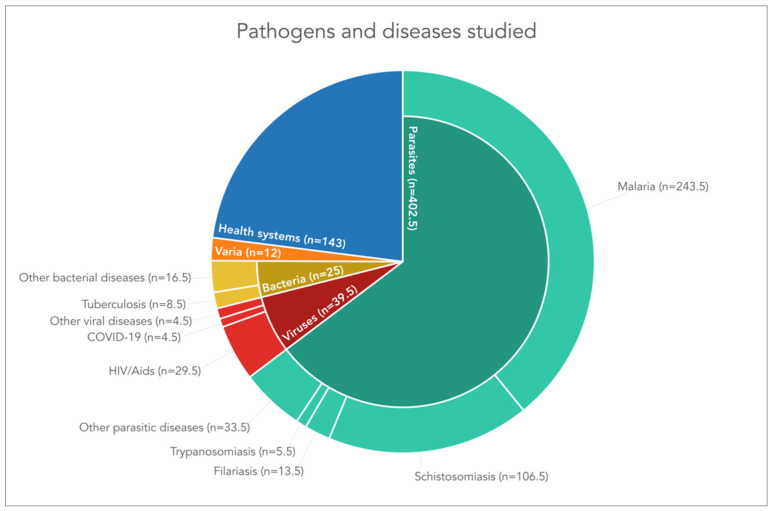
Diseases and pathogens studied, as revealed by content analysis of Marcel Tanner’s research output between 1978 and 2021. Note: the figure displays the types of pathogens in the inner pie chart and the accompanying diseases in the outer ring.

**Figure 6 diseases-10-00116-f006:**
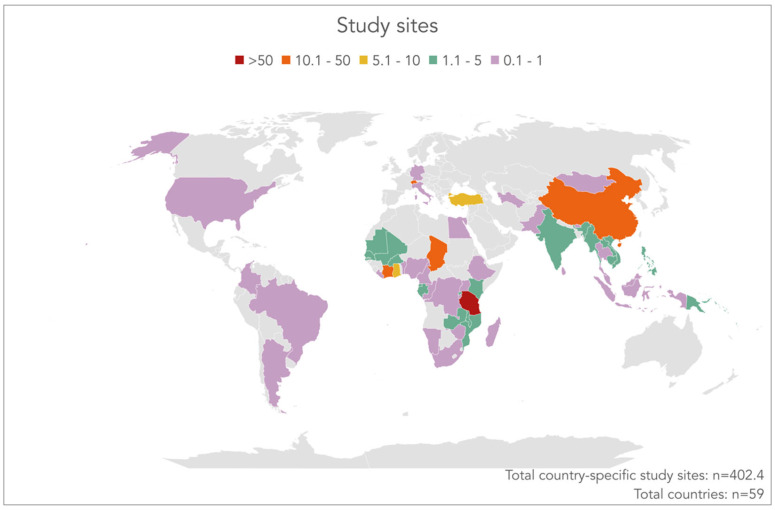
Country-specific study sites in Marcel Tanner’s publications between 1978 and 2021.

**Figure 7 diseases-10-00116-f007:**
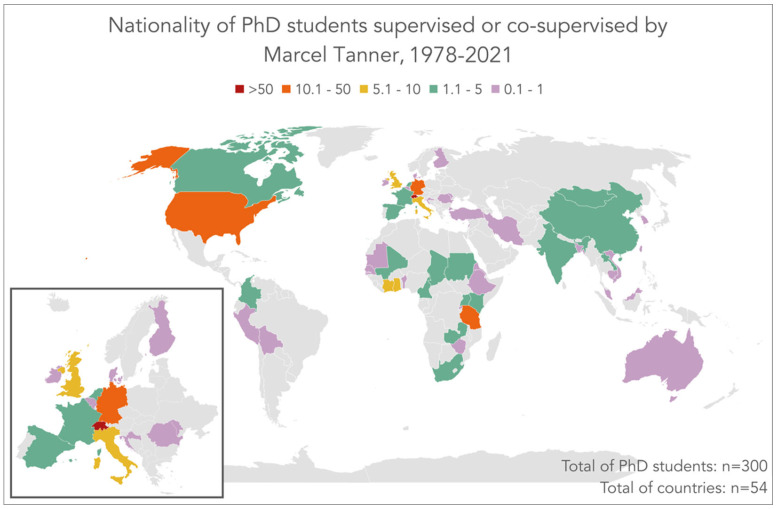
Nationality of Ph.D. students supervised or co-supervised by Marce Tanner between 1978 and 2021.

## Data Availability

Not applicable.

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
