# Peer review of "No Roots, No Fruits: Marcel Tanner’s Scholarly Contribution, Achievements in Capacity Building, and Impact in Global Health"

_diseases, 2022, doi:10.3390/diseases10040116_

Round 1

Reviewer 1 Report

This is interesting analysis of prof. Marcel Tanner's studies and activities in publications. I have only one specific comment.

Could the authors present Tanner's articles in context of kind of paper? I mean 622 articles divided to original works and review articles? Were there also clinical case reports?

Author Response

Dear reviewer

Thank you for your positive feedback and valuable comments!
Regarding your question:

  • We have added more information in our sub-chapters 2.1. and 3.1. that make it clear how many of the 622 articles are original papers, reviews, letters, editorials, book chapters and proceedings papers. This information has been missing so far.
  • We used Web of Science categories for our analysis, which did not include a category named "clinical case report". We could therefore not consider clinical case reports in our analysis.

Thank you again for pointing out the missing information regarding document types through your comment. 

Best,
Andrea Graf, Lukas Meier, Jürg Utzinger

Reviewer 2 Report

This manuscript is unique in that unlike an original article, a review or case report this is one that is outlining the career of a distinguished academic. The article shows the geographic impact of this researchers work. It is a manuscript that discusses the wide range of collaborative outreach over an illustrious career that making it both interesting and useful to other researchers who may follow in this career.

While the work is about an eminent researcher with special interest in parasitic disease the manuscript concentrate and all his work thus shows more of the collaborative effect that specifically homing in on a special  disease entity.

As a result of the above point specific research questions as are being recommended may not be applicable for this particular manuscript.

That being said, the manuscript offers an overall view of a researcher who studies a lot of rare tropical infections including parasitic disease. This provides a sterling example for young researched to emulate.

Author Response

Dear reviewer

Thank you for the positive feedback and your comments, which reflect very well our intentions with the article.
We have made some minor spelling corrections.

Best,
Andrea Graf, Lukas Meier, Jürg Utzinger